# Clinician Communication Training to Increase Human Papillomavirus Vaccination Uptake: A Systematic Review and Meta-Analysis

**DOI:** 10.3390/vaccines12060611

**Published:** 2024-06-03

**Authors:** Nutthaporn Chandeying, Therdpong Thongseiratch

**Affiliations:** 1Department of Obstetrics and Gynecology, Faculty of Medicine Vajira Hospital, Navamindradhiraj University, Bangkok 10300, Thailand; nutthaporn027@gmail.com; 2Department of Pediatrics, Faculty of Medicine, Prince of Songkla University, Songkhla 90110, Thailand

**Keywords:** HPV vaccination, clinician communication, adolescent immunization, vaccination strategies, healthcare training, presumptive communication

## Abstract

The battle against Human Papillomavirus (HPV)-related cancers is hindered by suboptimal vaccination rates, despite the proven efficacy and availability of vaccines. This systematic review and meta-analysis addressed this issue by evaluating the impact of clinician communication training on increasing HPV vaccination uptake among adolescents. From an initial pool of 3213 records, six randomized controlled trials involving 245,195 participants across the United States were rigorously selected and analyzed. Our findings indicated that clinician communication training could enhance vaccination uptake rates by an average of 5.2%. Specifically, presumptive communication strategies, which proactively assume a patient’s acceptance of vaccination, achieved a significant 9.1% increase in uptake, markedly outperforming the 2.3% increase observed with more passive conversational techniques. Moreover, interventions that incorporated audit and feedback processes were particularly impactful, boosting vaccination rates by 9.4%. The most striking results emerged from combining presumptive communication with audit and feedback, which propelled the effectiveness to an 11.4% increase in vaccination rates. These outcomes highlight the pivotal role of deliberate, targeted clinician–patient communication in improving health interventions. This study offers actionable insights for healthcare providers and policymakers to refine communication strategies, thus potentially maximizing HPV vaccination rates and mitigating the spread of HPV-related conditions.

## 1. Introduction

Cervical cancer, primarily caused by the Human Papillomavirus (HPV), continues to pose a significant global health threat [1]. However, HPV is also linked to other types of cancers such as anal, oropharyngeal, penile, vulvar, and vaginal cancers [2]. Together, these HPV-associated cancers contribute substantially to the global cancer burden. In 2020, there were approximately 604,127 new cases of cervical cancer and 341,831 deaths, highlighting the severity of HPV’s impact [1]. The development and widespread availability of HPV vaccines represent a major advancement in cancer prevention. These vaccines, whose development was inspired by Professor Harald zur Hausen’s Nobel Prize-winning discovery of HPV’s role in cervical cancer, have shown remarkable efficacy in clinical trials and real-world applications [3,4,5]. The passing of Professor zur Hausen last year adds a poignant note to his legacy, reminding us of the profound impact that dedicated scientific research can have on public health [6].

Despite the availability of effective vaccines, global HPV vaccination rates remain low, particularly in low- and middle-income countries (LMICs) [7]. For instance, a recent report highlighted that less than half of the countries in sub-Saharan Africa have incorporated HPV vaccination into their national immunization schedules [8]. The World Health Organization (WHO) has set an ambitious goal within its global strategy to eliminate cervical cancer, aiming to vaccinate at least 90% of girls by the age of 15 by 2030 [9]. However, as of 2019, only around 13% of girls in LMICs received the recommended doses of the HPV vaccine, significantly lagging behind the target [10,11]. This underscores a critical need to enhance efforts to increase vaccination coverage to meet the WHO’s elimination targets.

To improve HPV vaccination uptake, various strategies have been deployed, with healthcare providers playing a pivotal role [12]. Provider-based strategies such as educational interventions and reminders have shown promise in increasing vaccine uptake. A previous meta-analysis found significant improvements in HPV vaccine initiation, with an increase of 3.7% in the initiation rate and a 9.4% increase in the percentage of patients receiving the next needed dose when healthcare providers engage in personalized counseling [13]. Notably, these strategies include presumptive communication approaches, which are direct and assertive. Presumptive communication involves providers presenting vaccinations as a standard part of healthcare, implying that the patient will accept them without explicitly asking for consent. For example, a healthcare provider might say, “Today, we’ll be giving your 9-year-old child the HPV vaccine that can prevent six types of cancer”. This approach leverages the norm of compliance with medical recommendations to increase vaccination uptake [14,15].

Conversely, conversational communication techniques encompass methods such as motivational interviewing (MI), strong recommendations, and shared decision making, which involve more engagement and deliberation with patients. These methods focus on understanding the patient’s perspective, addressing their concerns, and guiding them towards making informed health decisions. For instance, MI engages patients in a goal-oriented, collaborative conversation to strengthen their motivation for change. A provider using MI might say, “I understand that you have concerns about the HPV vaccine. Can you tell me more about what worries you? Let’s talk through these concerns together and find a solution that works for you”. Strong recommendations explicitly emphasize the importance and urgency of vaccination, providing clear and direct guidance from the healthcare provider, such as, “I strongly recommend that your child receives the HPV vaccine today. It’s a critical step in protecting them against HPV-related cancers”. Shared decision making involves the patient actively in the decision process, acknowledging their role in making health choices and ensuring that their values and preferences are considered. An example sentence might be, “Let’s discuss the benefits and any concerns you may have about the HPV vaccine. I’ll provide you with all the information you need, and together, we can decide the best course of action for your child’s health” [14,15].

Although previous systematic reviews have delineated various communication strategies to improve HPV vaccination rates, they have primarily focused on the effects without integrating meta-analytical techniques to quantitatively assess outcomes across studies [15,16]. These reviews discuss the differences in communication strategies qualitatively, lacking rigorous comparative analysis. Additionally, there is a noticeable absence of research on the most effective methods for training clinicians to utilize these strategies effectively [13,14,15,16]. This oversight highlights a significant gap in our understanding of how best to equip healthcare providers to implement these interventions successfully.

This systematic review and meta-analysis seeks to fill these gaps by specifically focusing on randomized controlled trials (RCTs) that evaluate clinician communication training aimed at improving HPV vaccination uptake. The objectives of this study are twofold: (1) to conduct a meta-analysis to quantify the effectiveness of these strategies, and (2) to perform a subgroup analysis to determine which components of these strategies are most effective. By providing a clearer picture of how to effectively communicate about HPV vaccination, this study aims to contribute significantly to the global efforts to increase vaccination rates and, ultimately, help eliminate cervical cancer as a major public health concern.

## 2. Materials and Methods

The research protocol was proactively registered with PROSPERO, the International Prospective Register of Systematic Reviews and the Open Science Framework (OSF), accessible at https://osf.io/ze46q/ (accessed on 19 May 2024), to underscore our commitment to methodological rigor and transparency. Additionally, our methods and the reporting of results were strictly in line with the detailed recommendations provided in the Preferred Reporting Items for Systematic Reviews and Meta-Analyses (PRISMA) 2020 guidelines [17], as well as the methodological standards set forth in the Cochrane Handbook for Systematic Reviews of Interventions [18].

### 2.1. Search Strategy

We conducted a comprehensive literature search using PubMed, Scopus, Web of Science, and the Cochrane Library on 3 January 2024. The search strategy was designed to include terms related to ‘HPV vaccination’, ‘communication’, ‘healthcare provider’, and ‘training’. Keywords and MeSH terms were used in various combinations: (HPV OR ‘human papillomavirus’) AND (vaccin* OR immuni* OR ‘vaccine uptake’) AND (communicat* OR counsel* OR ‘health education’) AND (clinician* OR provider* OR ‘health personnel’). The complete search strategies are provided in the Appendix A. Filters were initially applied to restrict the search to studies published in English from January 2000 to December 2023. However, since the first HPV vaccine became available in 2006, we focused on studies published from January 2006 to December 2023.

### 2.2. Eligibility Criteria

Eligibility for inclusion in this study was limited to peer-reviewed articles written in English that conformed to the PICOS (Population, Intervention, Comparator, Outcome, Study Design) framework as follows:

Population (P): We included studies involving healthcare providers such as clinicians, nurses, and health educators who are directly involved in administering HPV vaccinations. We excluded studies that involve non-healthcare professionals or healthcare providers who are not directly involved in the vaccination process, such as administrative staff or those in non-clinical roles.

Intervention (I): This review specifically focuses on interventions that involve communication training for clinicians, aimed at increasing HPV vaccination uptake. The communication training encompasses a variety of strategies such as educational interventions, presumptive communication, and MI, all designed to enhance interactions between healthcare providers and patients. Our study includes both onsite training sessions and online training delivered via web platforms or webinars. Excluded from this review were passive communication methods, such as the distribution of educational materials like brochures or posters.

Comparator (C): Included studies must compare the effectiveness of communication strategies against standard practices that do not employ enhanced communication methods or utilize different types of communication interventions as comparators. We excluded studies where the comparators involved non-communication-based strategies, such as pharmacological interventions or structural changes within healthcare settings.

Outcomes (O): The primary outcomes of interest in this review are HPV vaccine initiation rates. The secondary outcomes considered include completion rates and the receipt of subsequent doses when initiation data are not available. Studies that do not directly report on these specific vaccination rates, or instead focus on indirect outcomes such as changes in knowledge, attitudes toward vaccination, readiness to vaccinate, general health outcomes, or non-specific educational metrics, were excluded from this review. This focus ensures that our analysis directly assesses the impact of communication training on tangible vaccination actions.

Study design (S): We included only RCTs in this review, as they provide the highest level of evidence for assessing the efficacy of interventions. We excluded non-randomized studies, observational studies, case reports, review articles, and qualitative studies to maintain the rigor and specificity of the evidence evaluated in this meta-analysis.

### 2.3. Study Selection

The reference lists of relevant systematic reviews and primary studies were reviewed. NC and TT independently conducted an initial screening of titles and abstracts using Rayyan (https://www.rayyan.ai/, accessed on 7 January 2024), a systematic review software, to identify studies potentially meeting the eligibility criteria. Subsequently, full-text articles of these potentially eligible studies were thoroughly evaluated for final inclusion by a research assistant along with NC and TT.

### 2.4. Data Extraction

This included general study characteristics (such as study design and duration), specific details about the interventions (such as the type of communication training and delivery method), characteristics of the study sample (including demographic information and setting), and relevant outcome data necessary for calculating effect sizes. For studies that reported both intention-to-treat and per-protocol analyses, intention-to-treat data were prioritized to maintain consistency and robustness in our analysis. In studies employing cluster sampling, the sample sizes were adjusted based on the reported design effect and intracluster correlation coefficients to accurately reflect this study’s impact [19]. 

All extracted data were systematically organized and recorded in Microsoft Excel. Data extraction and coding were performed by a research assistant, overseen by NC and TT, to ensure the accuracy and reliability of the data handling process. When necessary, authors of the studies were contacted to clarify or obtain additional data that were not available from the publications.

### 2.5. Quality Assessment

To ascertain the credibility of the cluster randomized trials included in our systematic review, we employed the revised Cochrane risk-of-bias tool for randomized trials (RoB 2) specifically tailored for cluster-randomized trials [18]. This comprehensive tool enabled us to assess bias across several domains: the randomization process, deviations from intended interventions, missing outcome data, measurement of the outcome, and selection of the reported result. Each domain was meticulously examined to determine the level of bias present, with judgments categorized as ‘low risk’, ‘some concerns’, or ‘high risk’. The evaluation of each domain was conducted independently by NC and TT to enhance objectivity, with any discrepancies resolved through discussion or consultation with a third reviewer. Additionally, to explore potential publication bias, we utilized funnel plots, which provided a visual assessment of the symmetry in the distribution of effect sizes, further validating the robustness of our meta-analytical findings.

### 2.6. Data Synthesis and Analysis

Data were synthesized quantitatively using meta-analysis methods where appropriate, employing the Comprehensive Meta-Analysis software version 4 [20] to facilitate statistical analysis. Effect sizes were calculated for vaccine initiation rates using random-effects models to account for variability between studies. We used the risk difference of vaccination uptake as the main effect measure, represented as the mean percentage increase in vaccination uptake. Subgroup analyses were specifically planned to examine the impact of different types of communication strategies. These subgroups were categorized based on the nature of the communication approaches used in the interventions, such as conversational strategies (including shared decision making and MI) and presumptive strategies (which assume patient agreement to vaccination without eliciting explicit consent). Additionally, we conducted further subgroup analyses to evaluate the effects of interventions that included audit and feedback methods as part of the post-training process compared to those that did not incorporate these additional methods. This approach allowed us to assess whether ongoing evaluation and feedback after initial communication training could further influence the effectiveness of the interventions.

To accommodate for potential variability across the included studies, we utilized a random-effects model, which is better suited for handling the expected heterogeneity. The extent of this heterogeneity was quantitatively assessed using the *I^2^* statistic, with cut-off values interpreted as follows: 0–40% may indicate low heterogeneity, 30–60% may indicate moderate heterogeneity, 50–90% may indicate substantial heterogeneity, and 75–100% may indicate considerable heterogeneity, as recommended by the Cochrane Handbook for Systematic Reviews of Interventions [18]. Statistical significance was indicated by a *p*-value of less than 0.05. Uncertainty was expressed using 95% confidence intervals. Results are graphically presented using forest plots to visually represent the effect sizes and their confidence intervals.

## 3. Results

Figure 1 presents the flow diagram of the study selection process. Initially, searches across various electronic databases yielded 3213 studies. After removing duplicates, 725 studies were further examined. Subsequently, 35 full-text articles were retrieved for detailed evaluation. Following a thorough review, studies that did not align with the inclusion criteria were excluded. Additionally, a study identified through alternative methods, such as website searches, Google Scholar searches, citation chasing, and references lists of existing systematic reviews, was added, culminating in a total of six studies being included.

Although our search strategy was not limited to studies conducted in the United States, all six studies that met our inclusion criteria were from the United States [21,22,23,24,25,26]. Table 1 shows that these six studies were all cluster RCTs that randomized interventions at the center level. They took place in diverse geographic locations, including urban centers such as Pittsburgh, Pennsylvania [26], and broader regions encompassing 19 states across the USA [25]. The settings varied among these studies, involving primary care, pediatric, and family medicine clinics. A significant total of 245,195 participants aged 11 to 17 years were enrolled, demonstrating extensive engagement with the target demographic for HPV vaccination within various American communities. All studies used control variables such as treatment as usual or waitlist control. Moreover, each study specifically measured the outcome of HPV vaccine initiation, which was the primary focus of this systematic review, ensuring consistency across the data analyzed for impacts on vaccination uptake.

### 3.1. Intervention Strategies across Studies

The meta-analysis reviewed a wide array of intervention strategies designed to enhance HPV vaccination rates among adolescents within various primary care settings. These interventions employed diverse communication techniques and technological aids to facilitate education and training for healthcare providers (Table 2).

The studies reviewed targeted a broad range of clinicians within primary care, pediatric, and family medicine settings across diverse geographical locations in the United States. Communication strategies were tailored to enhance HPV vaccination uptake effectively. For instance, Brewer et al. [21] contrasted announcements versus conversation techniques, Dempsey et al. [22] focused on MI and presumptive recommendations, and Gilkey et al. [24] emphasized cancer prevention and follow-up counseling.

The training characteristics varied significantly across the studies, with interventions ranging from didactic sessions to interactive online modules. Brewer et al. [21] used video vignettes and didactic instruction, providing a one-hour in-clinic training session. Dempsey et al. [22] integrated digital tools such as decision aids and fact sheets into their training, alongside a parent education website, in a 2½ h communication training session. Finney, Rutten et al. [23] also utilized practice-specific information sheets and websites as part of their training, delivered through online modules and ongoing audits. Gilkey et al. [24] employed video vignettes in their training, which included individual assessments and feedback over six months. Szilagyi et al. [25] used online modules combined with text message reminders, spread across three modules with weekly text messages. Zimmerman et al. [26] implemented a comprehensive approach with the web-based 4 Pillars™ Program, including biweekly telephone coaching and continuous engagement throughout this study.

The use of technology in these interventions varied widely, reflecting the evolving landscape of educational tools in healthcare. While Brewer et al. [21] did not specify any particular technological methods, Dempsey et al. [22] and Finney Rutten et al. [23] incorporated decision aids, websites, and practice-specific information sheets into their training. Gilkey et al. [24] reinforced training outcomes with videos, and Szilagyi et al. [25] utilized a combination of online modules and text message reminders to maintain engagement. Zimmerman et al. [26] used the web-based 4 Pillars™ Program, which provided continuous online program evaluations and feedback.

Three studies incorporated tracking or audit and feedback mechanisms to assess the effectiveness of the interventions and guide continuous improvement. Finney Rutten et al. [23] implemented ongoing audits, and Zimmerman et al. [26] utilized a web-based program that provided regular feedback on performance. Gilkey et al. [24] also included individual assessment and feedback over a six-month period. These feedback mechanisms are crucial for adapting and refining strategies based on real-time data, enhancing the overall impact of the interventions on vaccination rates.

### 3.2. Meta-Analysis

The meta-analysis employed a random-effects model to assess the efficacy of interventions on HPV vaccination rates, analyzing eight effect sizes derived from six studies (Figure 2). The pooled risk difference was 0.052, demonstrating that the interventions contributed to a 5.2% increase in vaccination uptake (95% CI, 1.9–8.5; *p* = 0.002). Notably, there was considerable heterogeneity among the studies (I^2^ = 77%; Q = 30.03; df = 7; *p* < 0.001), indicating variability in the intervention effects across different study settings.

### 3.3. Training Modalities Associated with Intervention Effectiveness in HPV Vaccination Uptake

Subgroup analyses, detailed in Table 3, focused on the comparative effectiveness of clinician communication training modalities. The meta-analysis assessed two primary communication strategies: conversation and presumptive. Presumptive communication was significantly more effective, increasing vaccination rates by 9.1% (95% CI, 6.3%–11.9%; *p* < 0.001), with moderate heterogeneity (I^2^ = 47%). In contrast, the conversation strategy led to a smaller increase of 2.3% (95% CI, 0.9%–3.7%; *p* = 0.001) and exhibited a slightly higher heterogeneity (I^2^ = 54%). The difference in effectiveness between these strategies was statistically significant (Q = 8.235; df = 1; *p* = 0.004).

The analysis also examined the effect of integrating audit and feedback mechanisms within the interventions. The inclusion of these components was associated with a notable increase in vaccination uptake of 9.4% (95% CI, 3.0%–15.9%; *p* = 0.001), although this was accompanied by high heterogeneity (I^2^ = 82%). In contrast, interventions lacking audit and feedback mechanisms showed a lesser effect of 2.4% (95% CI, 0.8%–3.9%; *p* = 0.004), with moderate heterogeneity (I^2^ = 42%). The impact of audit and feedback was significant (Q = 4.107; df = 1; *p* = 0.043).

The most substantial effects were observed with the combined strategy of presumptive communication alongside audit and feedback, which yielded an increase in vaccination rates of 11.4% (95% CI, 8.0%–14.8%; *p* = 0.001), with the lowest heterogeneity among the analyzed groups (I^2^ = 22%). Strategies that did not incorporate both elements had a significantly smaller effect size of 2.5% (95% CI, 1.1%–3.9%; *p* = 0.001), with low heterogeneity (I^2^ = 24%). The advantage of the combined strategy was marked and statistically significant (Q = 14.095; df = 1; *p* = 0.001).

### 3.4. Publication Bias

Visual inspection of the funnel plot (Figure 3) revealed no apparent signs of publication bias, supporting the reliability of the meta-analysis results. The distribution of effect sizes across the studies was fairly symmetrical, suggesting that there was no systematic skewing of results, due to publication bias. Most of the effect sizes fell well within the funnel, indicating a relatively uniform distribution. Even the few effect sizes that did fall outside the funnel appeared to do so symmetrically on both sides of the mean, further diminishing concerns regarding potential bias. This symmetry is crucial as it suggests that both smaller and larger studies contributed similarly to the overall analysis, and that the meta-analytical conclusions are likely to be robust across different study sizes and conditions.

### 3.5. Risk of Bias

In evaluating the risk of bias across six cluster RCTs, the majority of studies demonstrated a low risk of bias in the initial randomization process and measurement of outcomes, which are critical for the credibility of the results. However, some concerns were raised in a few domains. Notably, Gilkey et al. showed potential bias in the timing of participant identification and recruitment [24]. Deviations from intended interventions were a concern in the studies by Dempsey et al. and Finney Rutten et al. [22,23], while missing outcome data were noted in Dempsey et al., Szilagyi et al., and Zimmerman et al. [22,25,26]. Lastly, the possibility of selective reporting was indicated in Brewer et al., Finney Rutten et al., and Zimmerman et al. [21,23,26]. Despite these issues, the overall low risk of bias suggests the study findings are robust, though caution is advised in interpreting specific outcomes where biases were indicated (Figure 4).

## 4. Discussion

We identified six cluster RCTs that used clinician communication training to improve HPV vaccination uptake. These interventions involved both strategies, presumptive and conversational, in encouraging vaccination. Some interventions consisted of audit and feedback, whereas others did not have this approach. A previous systematic review identified several clinician communication trainings to improve HPV vaccination uptake but did not arrive at conclusive findings about the pooled effects [15]. To the best of our knowledge, this is the first study to focus on interventions conducted at the healthcare provider level and to estimate the pooled effects of clinician communication training on HPV vaccine uptake. Our study provides useful information on the effectiveness and feasibility of implementing clinician communication training to increase HPV vaccination coverage rates.

The findings from our meta-analysis, which demonstrated a 5.2% increase in HPV vaccination uptake due to targeted communication training interventions, offer a promising perspective on the potential of specific clinician education to improve vaccine rates. This increase is notably higher compared to the general provider-based interventions, which have been previously shown to improve vaccination rates by approximately 3.7% [13]. Such a difference underscores the effectiveness of tailored communication strategies over broader provider-based initiatives [27]. 

The significant role of providers’ communication in influencing vaccination uptake is well documented. For instance, research using the 2017 to 2018 National Immunization Survey-Teen highlighted that parental intent to initiate the HPV vaccine series was strongly associated with receiving communication from healthcare providers. Specifically, parents who received such recommendations were substantially more likely to initiate HPV vaccinations for their children, demonstrating the influence of provider engagement on vaccination decisions [28]. However, the substantial heterogeneity observed in our meta-analysis (*I*^2^ = 77%) suggests that the impact of communication training varies across different settings. This variation may stem from differences in implementation methods, population characteristics, or specific communication techniques used. Understanding these contextual factors is essential for refining intervention designs and achieving uniform improvements in vaccination rates across diverse settings. This leads us to focus on the results of our subgroup analysis, which further explore how specific aspects of these interventions influence their effectiveness.

Our subgroup analysis demonstrated that training presumptive communication strategies are substantially more effective—nearly four times—than training other conversational methods in enhancing HPV vaccination rates, with an increase of 9.1% compared to 2.3%. This significant finding from a meta-analysis validates the efficacy of presumptive approaches as more direct and standard practice in healthcare, likely reducing vaccination hesitancy more effectively than the more passive conversational methods [29]. However, the observed heterogeneity suggests that the impact of these strategies can vary across different settings, influenced by factors such as the implementation method. To address these variations, we conducted a subgroup analysis focusing on the role of audit and feedback mechanisms to see how they might further influence the effectiveness of communication training in boosting HPV vaccination uptake. The significant improvement in HPV vaccination rates found in this meta-analysis, particularly with the use of audit and feedback mechanisms (9.4% increase compared to 2.4%), supports the literature on enhancements in medical practices. Systematic reviews have consistently shown that providing systematic feedback to healthcare providers boosts their performance [13,27]. Furthermore, the notably robust effect of combining presumptive communication with audit and feedback strategies—resulting in an 11.4% increase—suggests a synergistic effect that tackles multiple barriers to vaccination effectively. This approach aligns with the CDC’s guidelines for comprehensive vaccination strategies, which advocate for multi-component interventions specifically tailored to overcome local challenges [30]. The low heterogeneity observed across studies highlights the potential for these combined strategies to be adopted universally as best practices for enhancing vaccination rates.

The findings from this meta-analysis underscore the necessity for targeted clinician communication training as a vital component of public health strategies aimed at increasing HPV vaccination rates [31]. The distinct effectiveness of presumptive communication techniques suggests that health policy should prioritize these methods in training programs. Moreover, integrating audit and feedback mechanisms could further enhance these effects, suggesting that policies should also support the infrastructure needed for continuous provider assessment and feedback. By incorporating these strategies into routine training for healthcare providers, health systems can significantly improve vaccination uptake, which is critical for reducing the burden of HPV-related diseases.

Almost all studies included in this meta-analysis incorporated some form of technology into their interventions, ranging from online modules and websites to text message reminders and video vignettes. Despite this widespread use of technology, there is still limited understanding of which specific technological methods are the most effective in enhancing communication strategies. Future research should focus on identifying which elements of communication training are most effective in different healthcare settings and populations. Since our findings revealed significant heterogeneity in the effectiveness of communication strategies, further studies should explore the reasons behind this variability. Additionally, research should assess the long-term impact of these interventions on vaccination rates and whether initial improvements are sustained over time. Investigating the cost-effectiveness of these interventions will also provide valuable insights for policymakers and healthcare administrators.

This study has some limitations. First, although our search strategy was not limited to studies conducted in the United States, all studies that met our inclusion criteria were conducted in the United States. This may limit the generalizability of the findings to healthcare settings in other countries, particularly in developing healthcare systems that may face unique challenges such as resource limitations and varying levels of healthcare infrastructure. Second, while the increase in vaccination rates is encouraging, the actual implementation of the interventions may encounter barriers such as resource constraints, provider resistance, or varying patient demographics that were not fully accounted for in this analysis. Third, the limited number of studies included in each subgroup analysis (requiring at least three studies per subgroup for robust analysis) may affect the precision and reliability of subgroup effect estimates. Fourth, the amalgamation of MI and strong recommendation techniques under the umbrella of conversation strategies may have nuanced differences in effectiveness that this study could not separately analyze. These factors could influence the observed effectiveness of the interventions and should be considered when interpreting the results and planning future research. Additionally, sources of heterogeneity such as variations in study design, participant demographics, and intervention implementation could have influenced the results. These variations might contribute to differences in observed effectiveness across studies. Future research should aim to include a larger number of studies to perform more robust subgroup analyses, which could provide more precise insights into the effectiveness of different communication strategies in various contexts. By addressing these sources of heterogeneity, future studies can help refine our understanding of the most effective interventions for increasing HPV vaccination uptake. Future research should aim to explore the applicability and effectiveness of these communication strategies in diverse healthcare settings, particularly in low- and middle-income countries, to ensure broader generalizability and impact.

## 5. Conclusions

This meta-analysis provides compelling evidence that clinician communication training, particularly when incorporating presumptive communication strategies and audit and feedback mechanisms, can significantly enhance HPV vaccination uptake. This reinforces the importance of focused training programs in public health initiatives and supports the integration of these strategies into routine clinical practice to improve vaccination coverage. As HPV-related cancers continue to impact public health, enhancing communication strategies among providers represents a crucial step towards achieving broader vaccination goals and ultimately reducing the incidence of these preventable diseases.

## Figures and Tables

**Figure 1 vaccines-12-00611-f001:**
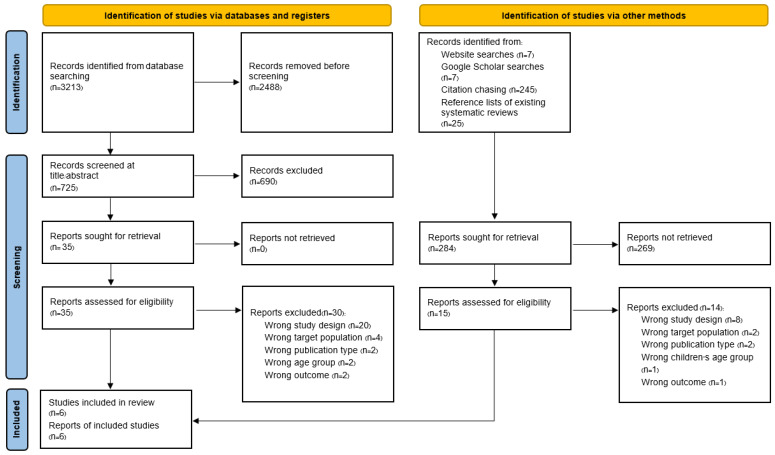
PRISMA 2020 flow diagram.

**Figure 2 vaccines-12-00611-f002:**
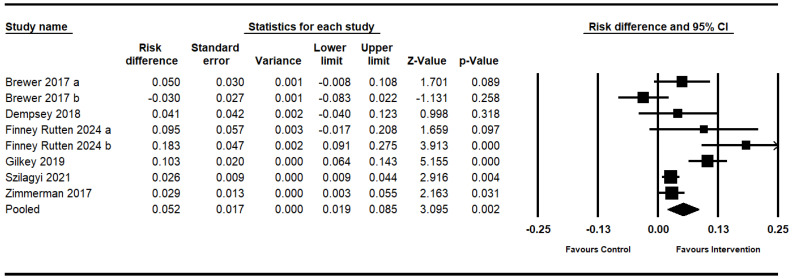
Forest plot of the effects on HPV vaccination uptake [21,22,23,24,25,26].

**Figure 3 vaccines-12-00611-f003:**
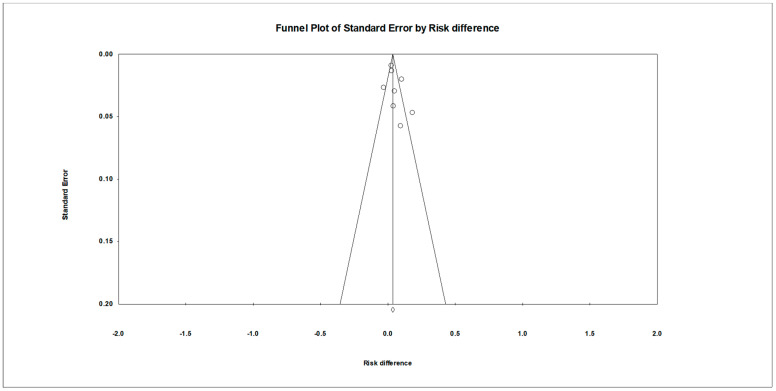
Funnel plot with symmetrical spread of effect sizes around the mean effect size.

**Figure 4 vaccines-12-00611-f004:**
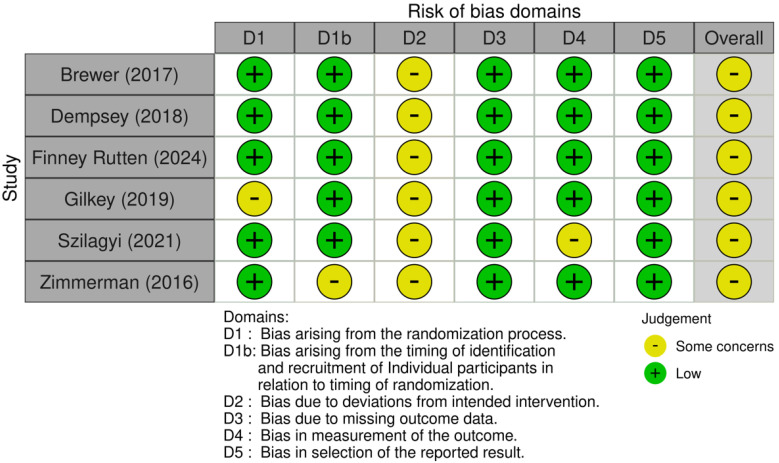
Risk-of-bias plot per study [21,22,23,24,25,26].

**Table 1 vaccines-12-00611-t001:** Basic characteristics across included studies.

Study	Location(s)	Study Design	Number of Clinics (Intervention/Control)	Number of Clinicians	Number of Participants	Age of Participants (years)	Brief Intervention Description	Control Comparison	Outcome (Intervention/Control)
Brewer et al., 2017 [21]	North Carolina, USA	Cluster RCT	Ped 22 Fam 7 (9/10/10) *	Not specifically mentioned	54,569	11–17	- Announcement training; - Conversation training.	No training	Initiation (42.0%/33.7%/41.2%) *, Completion (10.7%/9.2%/13.55%) *
Dempsey et al., 2018 [22]	Denver, Colorado, USA	Cluster RCT	Ped 12 Fam 4 (8/8)	188	43,132	11–17	5-component communication and educational intervention.	Standard of care	Initiation (42.9%/38.9%), Completion (72.4%/68.1%)
Finney Rutten et al., 2024 [23]	Southeastern Minnesota, USA	Stepped-wedge cluster RCT	Pri 6 (6/6/6) **	Not specifically mentioned	9242	11–12	- Professional audit/feedback; - Professional audit/feedback+ Parent reminders.	Usual care	Initiation (26.5%/28.7%/17.6%) **, Completion (39.8%/59.3%/36.0%) **
Gilkey et al., 2019 [24]	Fort Worth, Texas, USA	Cluster RCT	Ped 25 (13/12)	77	22,983	12–14	Repeated contacts and education; Feedback and incentives.	Wait-list control	Initiation (61.2%/50.9%)
Szilagyi et al., 2021 [25]	19 states, USA	Cluster RCT	Ped 48 (24/24)	365	104,438	11–17	Single Intervention: Online communication training and text message reinforcements.	Standard care	Initiation (32.1%/29.5%), subsequent dose (57.1%/56.5%)
Zimmerman et al., 2016 [26]	Pittsburgh, PA, USA	Cluster RCT	Pri 22 (11/11)	Not specifically mentioned	10,861	11–17	Single Intervention: 4 Pillars™ Practice Transformation Program	Usual care	Initiation (10.2%/7.3%), Completion (12.8%/12.7%)

Note: Ped = pediatric clinic, Fam = family medicine clinic, Pri = primary care clinic; * nine clinics for announcement training/10 clinics for conversation training/10 clinics for control; ** The numbers 6/6/6 indicate the allocation: clinics receiving Professional audit and feedback/clinics receiving Professional audit and feedback + Parent reminders/control group. *** Clinics receiving Professional audit and feedback/clinics receiving Professional audit and feedback + Parent reminders/control group.

**Table 2 vaccines-12-00611-t002:** Detailed comparison of interventions used in included studies.

Study	Target Clinicians	Communication Strategies Used	Training Character	Tech Use	Duration and Frequency	Tracking or Audit and Feedback Use
Brewer et al., 2017 [21]	Pediatric and family medicine clinicians	Announcement vs. conversation techniques	Didactic instruction and video vignettes	No specific tech mentioned	1 h in-clinic training	No
Dempsey et al., 2018 [22]	Pediatric and family medicine clinicians	Motivational interviewing, presumptive recommendations	Fact sheets, decision aids, website, disease images	Parent education website, decision aids	2½ hours of communication training	No
Finney Rutten et al., 2024 [23]	Primary care practices	Presumptive recommendations	Audit/feedback report	Use of practice-specific information sheets and websites	Online modules, ongoing audits	Yes, audit/feedback on performance
Gilkey et al., 2019 [24]	Pediatricians	Emphasizing cancer prevention, follow-up counseling	Didactic instruction, video vignettes	Videos used in training	1 h training, individual assessment and feedback over 6 months	Yes, individual assessment and feedback
Szilagyi et al., 2021 [25]	Pediatricians	Effective HPV recommendation, handling parental concerns	Online modules, text message reminders	Online modules, text messages	3 online modules, weekly text messages	No
Zimmerman et al., 2016 [26]	Primary care pediatric and family medicine practices	Office systems enhancements, motivating staff through champions	Education, coaching, web-based program	Web-based 4 Pillars™ Program	Biweekly telephone coaching, continuous throughout study	Yes, online program evaluation and provision of feedback

**Table 3 vaccines-12-00611-t003:** Training modalities associated with intervention effectiveness in HPV vaccination uptake.

Intervention Modality	Relative Effect Estimate (95% CI)	*p* Value	Number of Effect Sizes	*I* ^2^
1.Communication strategy	Q (df = 1) = 8.235	0.004		
Conversation	2.3% (0.9% to 3.7%)	0.001	3	54
Presumptive	9.1% (6.3% to 11.9%)	<0.001	5	47
2. Audit and feedback	Q (df = 1) = 107.4	0.043		
Audit and feedback	4.9% (0.3% to 1.9%)	0.001	4	82
No audit and feedback	2.4% (0.8% to 3.9%)	0.004	4	42
3. Combine strategy	Q (df = (1 = 95.14))	0.001		
Presumptive strategy with audit and feedback	11.4% (8.0% to 14.8%)	<0.001	3	22
The other interventions	2.5% (1.1% to 3.9%)	<0.001	5	24

## Data Availability

The data supporting the findings of this systematic review and meta-analysis are derived from publicly available articles. Detailed references to the original studies are provided within the manuscript. Data generated during the analysis are contained within this published article and its Appendix A. For access to these data, inquiries can be directed to the corresponding author, who will facilitate data requests as per applicable privacy and ethical guidelines.

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
