# Peer review of "Clinician Communication Training to Increase Human Papillomavirus Vaccination Uptake: A Systematic Review and Meta-Analysis"

_vaccines, 2024, doi:10.3390/vaccines12060611_

Round 1

Reviewer 1 Report

Comments and Suggestions for Authors

This meta-analysis might be helpful for developed countries where healthcare system is very advanced along with literacy rate as compare to developing countries. But I have doubt it might be fail in developing healthcare system. Anyway, authors did very through analysis which is impressive. 

minor comments:

In table1. authors should have abbreviate Ped and Fam.

Why authors did not mentioned the name of clinic like first two? they should. 

Author Response

Response to the Reviewer 1

This meta-analysis might be helpful for developed countries where healthcare system is very advanced along with literacy rate as compare to developing countries. But I have doubt it might be fail in developing healthcare system. Anyway, authors did very through analysis which is impressive.

minor comments:

In table1. authors should have abbreviate Ped and Fam.

Why authors did not mentioned the name of clinic like first two? they should.

Response to the Reviewer 1

General Comment:

We acknowledge your point regarding the applicability of our meta-analysis to different healthcare systems. While all included RCTs primarily focuses on developed countries, we believe the insights gained can still provide valuable guidance for improving healthcare practices globally. However, we will add a section discussing the limitations and applicability to developing healthcare systems to address this concern.

The revised discussion section:

This study has some limitations. First, the analysis was confined to studies conducted in the United States, which may limit the generalizability of the findings to healthcare settings in other countries, particularly in developing healthcare systems that may face unique challenges such as resource limitations and varying levels of healthcare infra-structure. Second, while the increase in vaccination rates is encouraging, the actual implementation of the interventions may encounter barriers such as resource constraints, provider resistance, or varying patient demographics that were not fully accounted for in this analysis. Third, the limited number of studies included in each subgroup analysis (requiring at least three studies per subgroup for robust analysis) may affect the precision and reliability of subgroup effect estimates. Fourth, the amalgamation of motivational interviewing and strong recommendation techniques under the umbrella of conversation strategies may have nuanced differences in effectiveness that this study could not separately analyze. These factors could influence the observed effectiveness of the interventions and should be considered when interpreting the results and planning future research. Future research should aim to explore the applicability and effectiveness of these communication strategies in diverse healthcare settings, particularly in low- and middle-income countries, to ensure broader generalizability and impact.

Minor Comments:

Abbreviations in Table 1:

We apologize for the oversight and agree that using abbreviations for clarity is beneficial. We have revised Table 1 to include the abbreviations: "Ped" for Pediatric clinic and "Fam" for Family medicine clinic. Additionally, we have included the abbreviation "Pri" for Primary care clinic.

Clinic Names:

Thank you for pointing out the inconsistency. We understand the importance of consistency in our presentation. We have included the names of the clinics for the first two rows and provided the abbreviation "Pri" for Primary care clinic. We have also ensured that this information is clearly stated and consistent throughout the table. 

Reviewer 2 Report

Comments and Suggestions for Authors

The manuscript shows good potential, but some information appears to require adjustments, specifically:

- Provide more detailed descriptions of the analyses conducted in the methods section. Which parameters were used in the Comprehensive Meta-Analysis software?

- I suggest making your dataset available for access.

- While the manuscript acknowledges some limitations, it would be beneficial to discuss in more depth how these limitations impact the generalization of the results.

- It would be helpful to expand the discussion on sources of heterogeneity and how they may have influenced the results. Additionally, a more robust subgroup analysis, possibly with a larger number of studies, could provide more precise insights into the effectiveness of interventions in different contexts.

- Offer a clearer distinction between the various communication strategies studied, such as motivational interviewing and presumptive recommendations.

Comments on the Quality of English Language

The manuscript is clear and only requires minor corrections that do not affect its clarity.

Author Response

Response to the Reviewer 2

The manuscript shows good potential, but some information appears to require adjustments, specifically:

Comment 1: Provide more detailed descriptions of the analyses conducted in the methods section. Which parameters were used in the Comprehensive Meta-Analysis software?

Response to Reviewer Comments:

Thank you for your valuable feedback. We appreciate your suggestion to provide more detailed descriptions of the analyses conducted. We will enhance the methods section to include specific parameters used in the Comprehensive Meta-Analysis software.

Revised Methods Section:

Data were synthesized quantitatively using meta-analysis methods where appropriate, employing the Comprehensive Meta-Analysis software [20] to facilitate statistical analysis. Effect sizes were calculated for vaccine initiation rates using random-effects models to account for variability between studies. We used the risk difference of vaccination uptake as the main effect measure, which is represented as the mean percentage increase in vaccination uptake.

Comment 2: I suggest making your dataset available for access.

Response to Reviewer Comments:

Thank you for your suggestion regarding data accessibility. We appreciate the importance of transparency and data sharing in research. We have made the dataset available for access through the Open Science Framework (OSF). The link to access the dataset is provided in the methods section of our manuscript.

Revised Methods Section:

The research protocol was proactively registered with PROSPERO, the International Prospective Register of Systematic Reviews and the Open Science Framework (OSF), accessible at https://osf.io/ ze46q/

Comment 3: While the manuscript acknowledges some limitations, it would be beneficial to discuss in more depth how these limitations impact the generalization of the results.

Response to Reviewer Comments:

Thank you for your insightful feedback. We agree that it is important to discuss in more depth how the limitations impact the generalization of the results. We have added the following text to the discussion section to address this:

Addition to Discussion Section:

This study has some limitations. First, the analysis was confined to studies conduct-ed in the United States, which may limit the generalizability of the findings to healthcare settings in other countries, particularly in developing healthcare systems that may face unique challenges such as resource limitations and varying levels of healthcare infra-structure.

Comment 4: It would be helpful to expand the discussion on sources of heterogeneity and how they may have influenced the results. Additionally, a more robust subgroup analysis, possibly with a larger number of studies, could provide more precise insights into the effectiveness of interventions in different contexts.

Response to Reviewer Comments:

Thank you for your valuable feedback. We agree that expanding the discussion on sources of heterogeneity and their influence on the results is important. Additionally, we acknowledge the potential benefits of conducting more robust subgroup analyses with a larger number of studies. We have added the following text to the discussion section to address these points:

Addition to Discussion Section:

Additionally, sources of heterogeneity such as variations in study design, participant demographics, and intervention implementation could have influenced the results. These variations might contribute to differences in observed effectiveness across studies. Future research should aim to include a larger number of studies to perform more robust sub-group analyses, which could provide more precise insights into the effectiveness of different communication strategies in various contexts. By addressing these sources of heterogeneity, future studies can help refine our understanding of the most effective interventions for increasing HPV vaccination uptake. Future research should aim to explore the applicability and effectiveness of these communication strategies in diverse healthcare settings, particularly in low- and middle-income countries, to ensure broader generalizability and impact.

Comment 5: Offer a clearer distinction between the various communication strategies studied, such as motivational interviewing and presumptive recommendations.

Response to Reviewer Comments:

Thank you for your valuable feedback. We agree that offering a clearer distinction between the various communication strategies studied, such as motivational interviewing and presumptive recommendations, is important. We have revised the introduction to provide these distinctions more clearly.

Revised Introduction Section:

Presumptive communication involves providers presenting vaccinations as a standard part of healthcare, implying that the patient will accept them without explicitly asking for consent. For example, a healthcare provider might say, "Today, we'll be giving your 9-year-old child the HPV vaccine that can prevent six types of cancer." This approach leverages the norm of compliance with medical recommendations to increase vaccination uptake [14,15].

Conversely, conversational communication techniques encompass methods such as motivational interviewing (MI), strong recommendations, and shared decision-making, which involve more engagement and deliberation with patients. These methods focus on understanding the patient's perspective, addressing their concerns, and guiding them to-wards making informed health decisions. For instance, MI engages patients in a goal-oriented, collaborative conversation to strengthen their motivation for change. A provider using MI might say, "I understand that you have concerns about the HPV vaccine. Can you tell me more about what worries you? Let's talk through these concerns together and find a solution that works for you." Strong recommendations explicitly emphasize the importance and urgency of vaccination, providing clear and direct guidance from the healthcare provider, such as, "I strongly recommend that your child receives the HPV vaccine today. It's a critical step in protecting them against HPV-related cancers." Shared decision-making involves the patient actively in the decision process, acknowledging their role in making health choices and ensuring that their values and preferences are considered. An example sentence might be, "Let's discuss the benefits and any concerns you may have about the HPV vaccine. I'll provide you with all the information you need, and together, we can decide the best course of action for your child's health" [14,15].

Reviewer 3 Report

Comments and Suggestions for Authors

This is an excellent systematic review and meta-analysis evaluating the methods that could enhance uptake of HPV vaccine. The manuscript is well written. However, some issues need to be addressed.

1. Line 87: Add the abbreviation (RCTs) as you used this term in other parts of the manuscript. And then remove the spell out from line 142.

2. Line 110: The first HPV vaccine became available in 2006. Considering that, why did the authors run their literature search from 2000? This sounds illogical, especially that no studies before 2006 were identified. I suggest that the authors edit their search window in the text to start from 2006. Also add a justification that this period was selected since the first HPV vaccine became available in 2006. No need to run the literature search again, just edit the text to avoid confusing the reader.

3. Line 124: You already used the abbreviation MI on line 66, so either use the abbreviation or remove it from lines 66 and 69.

4. Line 207: Please list or describe the alternative methods used.

5. Table 1: Please spell out the abbreviations "Ped" and "Fam" in a footnote under the table.

6. Table 1: What do I1 and I2 mean? Intervention 1 and intervention 2? If so, then it's better to remove these symbols and just list the interventions in bullet points. Also, for all the text in this column as well as the control comparison column, it would be helpful to add the number of participants in each group (n=?).

7. Table 1: In the outcome table, it would be better to add the % of participants who achieved each outcome.

8. Table 2: For the study by Finney Rutten, et al and under the column "Target Clinicians", it might be better to remove the numbers to ensure consistency with the other information in the same column. Also, the number of adolescents seems to be typed here by mistake since the column clinicians.

9. Line 315: Add the plural "s" to "RCT"

10. Line 318: Change to "others did not have this approach."

11. Lines 318-320: Please cite these previous systematic reviews.

12. Line 357: I suggest combining this paragraph with the previous one to maintain the flow of thought regarding findings related to audit and feedback.

13. In the discussion, could you please add if the utilization of online materials (websites or information sheets) had any additional effects of the outcomes of the studies that used them compared with the others that did not use them? Were certain tech methods better than others? This should be first mentioned in the results after Table 2 and then interpreted in the discussion.

Author Response

Response to the Reviewer 3

This is an excellent systematic review and meta-analysis evaluating the methods that could enhance uptake of HPV vaccine. The manuscript is well written. However, some issues need to be addressed.

  1. 1. Line 87: Add the abbreviation (RCTs) as you used this term in other parts of the manuscript. And then remove the spell out from line 142.

Response to Reviewer Comments:

Thank you for your positive feedback on our manuscript. We appreciate your suggestion and have made the necessary changes. We have added the abbreviation (RCTs) in line 87 and removed the spelled-out term from line 142.

  1. 2. Line 110: The first HPV vaccine became available in 2006. Considering that, why did the authors run their literature search from 2000? This sounds illogical, especially that no studies before 2006 were identified. I suggest that the authors edit their search window in the text to start from 2006. Also add a justification that this period was selected since the first HPV vaccine became available in 2006. No need to run the literature search again, just edit the text to avoid confusing the reader.

Response to Reviewer Comments:

Thank you for your insightful feedback. Although our initial search strategy covered the period from 2000 to 2023, we recognize that focusing on studies from 2006 onwards is more appropriate given that the first HPV vaccine became available in 2006. We have edited the text accordingly to avoid any confusion for the readers.

Revised Methods Section:

Filters were initially applied to restrict the search to studies published in English from January 2000 to December 2023. However, since the first HPV vaccine became available in 2006, we focused on studies published from January 2006 to December 2023.

  1. 3. Line 124: You already used the abbreviation MI on line 66, so either use the abbreviation or remove it from lines 66 and 69.

Response to Reviewer Comments:

Thank you for pointing out the inconsistency in the use of the abbreviation. We have revised the text to consistently use the abbreviation "MI" throughout the manuscript.

  1. 4. Line 207: Please list or describe the alternative methods used.

Response to Reviewer Comments:

Thank you for your suggestion. We have listed and described the alternative methods used in our study selection process.

Revised Results Section:

Additionally, a study identified through alternative methods, such as website searches, Google Scholar searches, citation chasing, and references lists of existing systematic re-views, was added, culminating in a total of six studies being included.

  1. 5. Table 1: Please spell out the abbreviations "Ped" and "Fam" in a footnote under the table.

Response to Reviewer Comments:

Thank you for your suggestion. We have added a footnote under Table 1 to spell out the abbreviations "Ped" and "Fam".

  1. 6. Table 1: What do I1 and I2 mean? Intervention 1 and intervention 2? If so, then it's better to remove these symbols and just list the interventions in bullet points. Also, for all the text in this column as well as the control comparison column, it would be helpful to add the number of participants in each group (n=?).

Response to Reviewer Comments:

Thank you for your valuable feedback. We have revised the table to list the interventions in bullet points and provided the number of clinics for each group, as all included studies are cluster RCTs. We did not provide the number of participants for each group because the randomization process was conducted at the clinic level, not the individual participant level.

  1. 7. Table 1: In the outcome table, it would be better to add the % of participants who achieved each outcome.

Response to Reviewer Comments:

Thank you for your valuable feedback. We have revised the table to include the percentage of participants who achieved each outcome.

  1. 8. Table 2: For the study by Finney Rutten, et al and under the column "Target Clinicians", it might be better to remove the numbers to ensure consistency with the other information in the same column. Also, the number of adolescents seems to be typed here by mistake since the column clinicians.

Response to Reviewer Comments:

Thank you for your valuable feedback. We have removed the numbers under the "Target Clinicians" column for the study by Finney Rutten et al. to ensure consistency with the other information in the same column.

  1. 9. Line 315: Add the plural "s" to "RCT"

Response to Reviewer Comments:

Thank you for your valuable feedback. We have added the plural "s" to "RCT" to ensure proper grammar.

  1. 10. Line 318: Change to "others did not have this approach."

Response to Reviewer Comments:

Thank you for your valuable feedback. We have revised the text as suggested.

  1. 11. Lines 318-320: Please cite these previous systematic reviews.

Response to Reviewer Comments:

Thank you for your valuable feedback. We have added citations to previous systematic reviews in the revised text.

  1. 12. Line 357: I suggest combining this paragraph with the previous one to maintain the flow of thought regarding findings related to audit and feedback.

Response to Reviewer Comments:

Thank you for your valuable feedback. We have combined the paragraph with the previous one to maintain the flow of thought regarding findings related to audit and feedback.

Revised Discussion:

To address these variations, we conducted a subgroup analysis focusing on the role of audit and feedback mechanisms to see how they might further influence the effectiveness of communication training in boosting HPV vaccination uptake. The significant improvement in HPV vaccination rates found in this meta-analysis, particularly with the use of audit and feedback mechanisms (9.4% increase compared to 2.4%), supports existing literature on enhancements in medical practices.

  1. 13. In the discussion, could you please add if the utilization of online materials (websites or information sheets) had any additional effects of the outcomes of the studies that used them compared with the others that did not use them? Were certain tech methods better than others? This should be first mentioned in the results after Table 2 and then interpreted in the discussion.

Response to Reviewer Comments:

Thank you for your valuable feedback. We recognize the importance of discussing the impact of utilizing online materials and other technologies on the outcomes of the studies. Although only Brewer et al., 2017 did not mention specific technologies, while other studies used various technological methods, we could not conduct a subgroup analysis for the meta-analysis. However, we will add relevant text in the results and discussion sections.

Revised Results:

The use of technology in these interventions varied widely, reflecting the evolving landscape of educational tools in healthcare. While Brewer et al. [21] did not specify any particular technological methods, Dempsey et al. [22] and Finney Rutten et al. [23] incor-porated decision aids, websites, and practice-specific information sheets into their train-ing. Gilkey et al. [24] reinforced training outcomes with videos, and Szilagyi et al. [25] uti-lized a combination of online modules and text message reminders to maintain engage-ment. Zimmerman et al. [26] used the web-based 4 PillarsProgram, which provided continuous online program evaluations and feedback.

Revised Discussion:

Almost all studies included in this meta-analysis incorporated some form of technology into their interventions, ranging from online modules and websites to text message reminders and video vignettes. Despite this widespread use of technology, there is still limited understanding of which specific technological methods are the most effective in enhancing communication strategies. Future research should focus on identifying which elements of communication training are most effective in different healthcare settings and populations. Since our findings revealed significant heterogeneity in the effectiveness of communication strategies, further studies should explore the reasons behind this variability. Additionally, research should assess the long-term impact of these interventions on vaccination rates and whether initial improvements are sustained over time. Investigating the cost-effectiveness of these interventions will also provide valuable insights for policy-makers and healthcare administrators.

Reviewer 4 Report

Comments and Suggestions for Authors

Dear Authors,

Your article presents very important public health information. However I have a comment that might deteriorate the siginificance of your results.

Page 3, line 108. You described the keywords for searching relevant papers. However immuniZation is correct in both British and American English, but you missed the fact that immuniSation is the word in Australian English and probably in some other English-speaking countries. That is a plausible explanation why did you find only trials performed in the Unitetd States.

I would suggest to describe this within the Limitations section. ImmuniZation is the MeSH term and I hope no article with a keyword immuniSation is missed. You can also perform an additional search (immunisation as a key word together with the other words you already used) and check if there are any missed articles that match your criteria.

It is the Editor‘s desicion what to propose for your article.

Author Response

Response to the Reviewer 4

Your article presents very important public health information. However I have a comment that might deteriorate the siginificance of your results.

Page 3, line 108. You described the keywords for searching relevant papers. However immuniZation is correct in both British and American English, but you missed the fact that immuniSation is the word in Australian English and probably in some other English-speaking countries. That is a plausible explanation why did you find only trials performed in the Unitetd States.

I would suggest to describe this within the Limitations section. ImmuniZation is the MeSH term and I hope no article with a keyword immuniSation is missed. You can also perform an additional search (immunisation as a key word together with the other words you already used) and check if there are any missed articles that match your criteria.

Response to Reviewer 4:

Thank you for your insightful comment and for highlighting the importance of ensuring comprehensive keyword coverage in our search strategy. We would like to clarify that we included studies using both the "immunization" and "immunisation" spellings, as we employed the search term "immuni*" to capture both variations. Additionally, we have corrected a typo in our manuscript where "immuniz*" should have been "immuni*".

Revised Text:

We conducted a comprehensive literature search using PubMed, Scopus, Web of Science, and the Cochrane Library. The search strategy was designed to include terms related to 'HPV vaccination,' 'communication,' 'healthcare provider,' and 'training.' Keywords and MeSH terms were used in various combinations: (HPV OR 'human papillomavirus') AND (vaccin* OR immuni* OR 'vaccine uptake') AND (communicat* OR counsel* OR 'health education') AND (clinician* OR provider* OR 'health personnel'). The complete search strategies are provided in the supplementary material.

It is the Editor‘s desicion what to propose for your article.

Round 2

Reviewer 2 Report

Comments and Suggestions for Authors

O texto foi significativamente melhorado. Isso torna as informações mais organizadas.

Comments on the Quality of English Language

Língua inglesa bem. Nenhum problema sério detectado

Author Response

Dear Reviewer,

Thank you for your valuable feedback and for your positive assessment of our manuscript.

We appreciate your comments highlighting that the text has been significantly improved and that the information is now more organized. Additionally, we are pleased to hear that the quality of the English language is well-regarded, with no serious issues detected.

We will continue to ensure the highest standards in our writing and presentation. Thank you once again for your constructive comments and support.

Best regards,

Reviewer 3 Report

Comments and Suggestions for Authors

I appreciate the efforts made by the authors to revise their manuscript. However, only some minor issues occurred during the revision.

1. Line 115: "HPV" s repeated.

2. Lines 137, 202 and 257: Remove the spell out "motivational reviewing" and just keep the abbreviation MI.

3. Lines 125 and 352: "RCTs" is repeated.

4. Table 2: The word "audit" is repeated under "Training character" column.

5. Line 262: Add a space before "providing"

6. Line 263: Add a space before "such as"

7. Line 355: "others" is repeated. Also is line, since you cited only one systematic review, then change the sentence to "A previous systematic review..."

Author Response

Dear Reviewer,

Thank you for your valuable feedback and for acknowledging the efforts we have made in revising our manuscript. We have addressed the minor issues you pointed out as follows:

  1. Line 115: We have carefully reviewed the entire manuscript and did not find the repetition of "HPV" on line 115 or elsewhere. It is possible that this was a technical issue in the previous manuscript file.

  2. Lines 137, 202, and 257: We have removed the spell out "motivational interviewing" and retained only the abbreviation "MI" as requested.

  3. Lines 125 and 352: The repetition of "RCTs" has been corrected.

  4. Table 2: The repetition of "audit" under the "Training character" column has been corrected.

  5. Line 262: A space has been added before "providing."

  6. Line 263: A space has been added before "such as."

  7. Line 355: The repetition of "others" has been corrected. Additionally, since only one systematic review was cited, we have changed the sentence to "A previous systematic review..."

We found that all repetitions or space issues may appear in some versions of Word software; however, we have corrected these issues and verified that they are not present in almost all Word versions. If you still encounter any repetition or space problems, please check with the PDF file of our manuscript for confirmation.

We appreciate your thorough review and believe that these changes have improved the clarity of our manuscript. Thank you once again for your constructive comments.

Best regards,
Therdpong Thongseiratch, MD, MSc 

Reviewer 4 Report

Comments and Suggestions for Authors

Dear Authors, 

Thank you for clarifying the issue.

Author Response

Thank you for your constructive comments and for your positive feedback. We believe that these changes have improved the clarity and quality of our manuscript.